# Metaplastic Carcinoma of the Breast: Case Series of a Single Institute and Review of the Literature

**DOI:** 10.3390/medsci11020035

**Published:** 2023-05-19

**Authors:** Alkistis Papatheodoridi, Eleni Papamattheou, Spyridon Marinopoulos, Ioannis Ntanasis-Stathopoulos, Constantine Dimitrakakis, Aris Giannos, Maria Kaparelou, Michalis Liontos, Meletios-Athanasios Dimopoulos, Flora Zagouri

**Affiliations:** 1Department of Clinical Therapeutics, Medical School of National and Kapodistrian University of Athens, “Alexandra” General Hospital of Athens, 115 28 Athens, Greece; 2Department of Physiology, Medical School of National and Kapodistrian University of Athens, 115 28 Athens, Greece; 31st Department of Obstetrics & Gynecology, “Alexandra” Hospital, Medical School, University of Athens, 115 28 Athens, Greece

**Keywords:** breast cancer, metaplastic carcinoma, triple-negative breast cancer

## Abstract

Metaplastic carcinoma of the breast (MpBC) is a very rare and aggressive type of breast cancer. Data focusing on MpBC are limited. The aim of this study was to describe the clinicopathological features of MpBC and evaluate the prognosis of patients with MpBC. Eligible articles about MpBC were identified by searching CASES SERIES gov and the MEDLINE bibliographic database for the period of 1 January 2010 to 1 June 2021 with the keywords metaplastic breast cancer, mammary gland cancer, neoplasm, tumor, and metaplastic carcinoma. In this study, we also report 46 cases of MpBC stemming from our hospital. Survival rates, clinical behavior, and pathological characteristics were analyzed. Data from 205 patients were included for analysis. The mean age at diagnosis was 55 (14.7) years. The TNM stage at diagnosis was mostly stage II (58.5%) and most tumors were triple negative. The median overall survival was 66 (12–118) months, and the median disease-free survival was 56.8 (11–102) months. Multivariate Cox regression analysis revealed that surgical treatment was associated with decreased risk of death (hazard ratio 0.11, 95% confidence interval 0.02–0.54, *p* = 0.01) while advanced TNM stage was associated with increased risk of death (hazard ratio 1.5, 95% confidence interval 1.04–2.28, *p* = 0.03). Our results revealed that surgical treatment and TNM stage were the only independent risk factors related to patients’ overall survival.

## 1. Introduction

Breast cancer is the most common cancer and most common cancer-related cause of death in women worldwide [1]. During the last decades, screening programs for breast cancer have become widely available to a large part of the population. At the same time, subtypes of breast cancer have been recognized mainly based on histological type, hormone receptor expression, and human epidermal growth factor receptor 2 (Her2) amplification. Different subtypes have a distinct prognosis and response to treatment, underlying the heterogeneity of the disease [2]. Concurrently, novel specified agents have been developed, altering the landscape of the standard of care for breast cancer and allowing us to offer various individualized treatment options to our patients [3,4,5]. However, even if the mortality rates of breast cancer tend to decline, it remains a major health problem, especially for women. Reduced access to the healthcare facilities due to the COVID-19 pandemic also resulted in delays in diagnosis and treatment, which may lead to a short-term drop in cancer incidence followed by an increase in late-stage disease and, ultimately, increased mortality [1].

Great progress has been made in the treatment of invasive ductal carcinoma of the breast, however rare subtypes of breast cancer have not been adequately studied. Metaplastic breast cancer (MpBC) is an uncommon histological subtype of breast cancer, which was officially recognized as a distinct pathological diagnosis in 2000. It constitutes approximately 0.2–5% of breast cancer cases [6]. This rare type of breast cancer is a general term referring to a heterogeneous group of neoplasms characterized by an intimate admixture of adenocarcinoma with dominant areas of spindle cell, squamous, and/or mesenchymal differentiation [6,7]. Histopathologically, it consists of various components, including poorly differentiated ductal, sarcomatous, and other epithelial elements [7].

Several studies and case reports exist in the current literature describing patient symptoms and clinicopathologic parameters [7,8]. Generally, MpBCs are considered as more aggressive with worse clinical outcomes even when compared to triple-negative breast cancer; however, data about treatment options especially for this type of breast cancer as well as treatment outcomes are limited [8,9].

In this systematic review, we aimed to incorporate available data from the literature in order to evaluate the characteristics and prognosis of patients with MpBC. Moreover, we provide a synopsis of the existing literature about the pathogenesis, molecular features, and clinical characteristics of metaplastic carcinoma, as well as treatment options and prognosis.

## 2. Materials and Methods

Our systematic review was conducted in accordance with the PRISMA guidelines [10]. All eligible articles were identified by searching CASES SERIES gov and the MEDLINE bibliographic database for the period of 1 January 2010 to 1 June 2021. The search strategy consisted of the following algorithm: ((metaplastic) AND (“breast” OR “mammary gland”) AND (“cancer” OR “neoplasm” OR “tumour” OR “tumor” OR “carcinoma”)). Moreover, in order to further investigate potentially eligible papers, we carefully examined all of the references cited in the reviews and eligible articles identified in our search. All prospective and retrospective studies as well as case reports that included patients with MpBC were eligible for our analysis, whereas reviews were ineligible. The following data were collected from all included studies: age, TNM stage, immunohistochemical characteristics of tumor (ER, PR, HER2, ki67%), treatment approach (type of surgery, neoadjuvant or adjuvant treatment), and data related to the follow-up of the patients (overall survival, disease-free survival). Cases with unknown TNM staging were excluded.

In the current study, we also report 46 cases of metaplastic breast cancer stemming from the Department of Clinical Therapeutics, Alexandra General Hospital, School of Medicine, National and Kapodistrian University of Athens, Greece.

Our database included the following information: histological type; TNM stage; estrogen receptor (ER), progesterone receptor (PR), HER2, and Ki67 status; age; treatment with surgery, chemotherapy in either a neoadjuvant, adjuvant, or metastatic setting, and adjuvant radiotherapy; patient outcomes regarding recurrence and overall survival; as well as the first author’s name and year of publication for each study.

The study protocol was approved by the Institutional Review Board of Alexandra Hospital, Medical School of Athens, Greece. Written informed consent was obtained from all individual participants included in the study for the anonymous use of their medical data and the study was conducted in accordance with the Declaration of Helsinki.

### 2.1. Follow-Up and Definitions

Disease-free survival (DFS) was defined as the interval (in months) between the date of diagnosis and the date of recurrence, death, or censored at last follow-up, whichever occurred first. Overall survival (OS) (in months) was defined as the interval between the date of diagnosis and the date of death or censored at last contact.

### 2.2. Statistical Analysis

Statistical analysis was performed using the statistical package IBM SPSS Statistics for windows, version 26 (IBM Corp., Armonk, NY, USA). Parametric variables were presented by their mean values ± standard deviation (SD). Nonparametric variables were presented as median values [interquartile range (IQR)]. The Kolmogorov–Smirnov goodness-of-fit test as used to assess the distribution of each variable. Categorical variables were summarized as frequencies and percentages. Kaplan–Meier curves were used for the estimation of the cumulative probability of death or recurrence. Independent prognostic factors of risk of death or recurrence (age; TNM stage; ER, PR, HER2, and ki-67 expression; surgery, chemotherapy, and radiation treatment) were identified using univariable and multivariable Cox proportional hazards models and presented as hazard ratios (HR) and 95% confidence intervals (CI) along with corresponding *p* values. For multivariate Cox regression analysis, we included all statistically significant variables in univariate analysis and potentially biologically important variables, namely age and TNM stage. A *p*-value of <0.05 was considered statistically significant.

## 3. Results

### 3.1. Search Strategy

Two independent investigators, A.P. and E.P., working independently, extracted data from each eligible study based on the title, abstract, and full text utilizing a predefined form. Our search strategy of the bibliographic databases recruited 997 articles, 856 of which were considered irrelevant. Out of these, 39 of the remaining studies did not contain information about TNM staging, while 38 studies did not contain patients’ individual data. Therefore, these 77 articles were excluded. Overall, 65 studies [11,12,13,14,15,16,17,18,19,20,21,22,23,24,25,26,27,28,29,30,31,32,33,34,35,36,37,38,39,40,41,42,43,44,45,46,47,48,49,50,51,52,53,54,55,56,57,58,59,60,61,62,63,64,65,66,67,68,69,70,71,72,73,74,75] including 159 patients with MpBC were included in our analysis, as shown in Figure 1.

### 3.2. Baseline Characteristics

The baseline characteristics of the 205 patients, including 159 patients recruited from our search and 46 patients treated in our hospital, are presented in Table 1. Baseline characteristics of the 46 patients treated in our institution are presented in the Appendix A. The mean age at diagnosis was 55 (±14.7 SD) years, as shown in Table 1. The TNM stage at diagnosis was more often stage II (58.5%) and the majority of tumors did not express ER (89.4%), PR (90%), or Her2 (91.1%). Ki67 expression was found to be high (>15%) in the majority of cases (68.4%) [2]. The main histological types were low-grade adenosquamous, squamous, spindle cell carcinoma, carcinosarcoma, matrix-producing, and metaplastic carcinoma with mesenchymal or osseous differentiation. Invasive ductal carcinoma of the breast often coexisted. Parameters such as distant relapse, disease-free survival, or overall survival were not always evaluated in published studies. In our institution, 21 (46.5%) cases of mixed metaplastic cancer, 10 (21.7%) cases of squamous cell carcinoma, 2 (4.3%) cases of spindle cell carcinoma, and 4 (8.7%) cases of matrix-producing carcinoma were treated. A total of 33 (72%) of our patients presented with TNM stage II at diagnosis, 8 (17.4%) patients presented with TNM stage I at diagnosis, 4 (8.7%) patients presented with TNM stage III, while only 1 patient presented with metastatic disease at diagnosis. Almost all patients underwent surgery and most of them were treated with systemic adjuvant chemotherapy, while more than 50% also received adjuvant radiotherapy.

### 3.3. Overall Survival

Median overall survival was calculated using Kaplan–Meier curves at 66 [12–118] months, as shown in Figure 2. Similarly, overall survival in every TNM stage was calculated using Kaplan–Meier curves, as shown in Figure 3. The median overall survival was 69 [33–105] months for stage I disease, 103 [38.25] months for stage II disease, 38.5 [1–84] months for stage III disease, and 19 [6–32] months for metastatic disease, as shown in Figure 3. Univariate Cox regression analysis for patients’ overall survival is presented in Table 2. Advanced TNM stage was associated with increased probability of death (HR 1.6, 95% CI 1.13–2.36, *p* = 0.01) while surgical treatment was associated with decreased risk of death (RH 0.06, 95% CI 0.02–0.2, *p* < 0.001). No other risk factor was significantly related to patients’ survival. Multivariate Cox regression analysis for age, TNM stage at diagnosis, and surgical treatment revealed that both TNM stage (HR 1.5, 95%CI 1.04–2.28 *p* = 0.03) and surgical treatment (HR 0.11, 95%CI 0.02–0.54 *p* = 0.01) were independent risk factors for patients’ survival, as shown in Table 2.

### 3.4. Disease-Free Survival

Similarly, the median disease-free survival, as estimated by Kaplan–Meier curves, was 56.8 [11–102] months, as presented in Figure 4. Univariate Cox regression analysis revealed that treatment with chemotherapy (HR 0.45, 95%CI 0.21–0.94 *p* = 0.03) and surgical treatment (HR 0.05, 95%CI 0.01–0.23 *p* < 0.001) were associated with decreased risk of disease recurrence. However, multivariate Cox regression analysis showed that surgical treatment was the only factor with a moderate tendency to decrease the risk of recurrence, which was at the borderline of statistical significance, as shown in Table 3.

## 4. Discussion

Metaplastic breast carcinoma is a rare subtype of breast cancer that affects only a small proportion of breast cancer patients [76]. Thus, no clinical trials exist for this type of cancer in particular. Therefore, according to international guidelines, treatment options for MpBC are similar to those of invasive ductal or lobular carcinoma of the breast, including: surgery (either mastectomy or breast-conserving lumpectomy), chemotherapy in an adjuvant or neoadjuvant setting, and radiation, depending on the hormonal receptor status and TNM stage at diagnosis [77]. However, MpBC is unique in terms of its histopathological and clinical features and presents a more aggressive course of disease than ductal or lobular carcinoma. Hence, appropriate management remains to be clarified as only limited data exist describing MpBC’s response to treatment.

### 4.1. Classification of MpBC

MpBC is a heterogeneous disease with different subgroups concerning histogenesis, biology, and prognosis [5]. According to the 2012 World Health Organization classification [78], MpBCs are divided into low-grade adenosquamous carcinoma, fibromatosis-like metaplastic carcinoma, squamous cell carcinoma, spindle cell carcinoma, metaplastic carcinoma with mesenchymal differentiation, myoepithelial carcinoma, and mixed metaplastic carcinoma. However, the classification of different subtypes of MpBC can be quite challenging. Several studies suggest that squamous cell carcinoma is the most commonly encountered subtype of MpBC [9]. The mixed type includes carcinoma with chondroid metaplasia, carcinoma with osseous metaplasia, and carcinosarcoma [31], while the sarcoma-like component may look like a malignant fibrous histiocytoma, chondrosarcoma, osteosarcoma, rhabdomyosarcoma, or a mixture of these [14]. Subcategorization of different subtypes of MpBC is crucial as prognosis varies between subtypes [35]. Yamaguchi et al. studied the prognosis of 53 cases of distinct subtypes of MpBC and showed that patients with high-grade spindle carcinoma or squamous carcinoma were at higher risk of recurrence and developing distant metastasis compared to patients with other MpBC subgroups [79]. In our study, patients of Alexandra Hospital with mixed or squamous carcinoma had local or distant relapse more often than patients with other MpBC subgroups.

### 4.2. Carcinogenesis

The histopathogenesis of this rare malignancy is unclear, as existing data remain limited. Several theories regarding the development of MpBCs have been proposed. MpBCs may derive from conventional mammary adenocarcinomas, which undergo metaplasis in a non-glandular growth pattern characterized by transepithelial differentiation and epithelial–mesenchymal phenotypic transition, following molecular genetic alterations associated with upregulation or downregulation of epithelial phenotypes [6,19]. Epithelial-to-mesenchymal transition could explain the more aggressive clinical course of MpBCs, as it is related to tumor invasion, migration, and metastasis. [6] Moreover, other studies suggest that malignant growth of intrinsic epidermal elements and metaplasis of breast parenchyma may lead to metaplastic breast carcinogenesis [48,65].

### 4.3. Genetic Mutations on MpBC

It is of interest that several genetic factors and pathways have been recently related to MpBC development. Molecular pathways including overexpression of CD44 (+) and yes-associated protein, which is associated with epithelial-to-mesenchymal transition, seem to play an important part in metaplastic breast carcinogenesis [15,23,41]. However, the expression of molecular markers related to carcinogenesis seems to depend on the histological subtype. More specifically, aberrant expression of Snail, a transcription factor that downregulates epithelial genes and is mostly observed in metaplastic carcinomas with chondroid difference, leads to changes in epithelial architecture, induction of epithelial-to-mesenchymal transition, and increases the risk of breast carcinogenesis and metastasis [23].

In addition, the genetic profile and landscape of gene mutations in breast cancer have been extensively studied in the past few years. As far as MpBC is concerned, several mutations have been identified. For example, p53, cyclin-dependent kinase inhibitor 2A, and epidermal growth factor genes play significant roles in cell cycle disruption and carcinogenesis, leading to a more aggressive phenotype and drug resistance in metaplastic carcinomas [6,15]. Somatic mutations in genes such as p53, phosphatidylinositol-4,5-bisphosphate 3-kinase catalytic subunit alpha (PIK3CA), breast cancer gene (BRCA), DNA topoisomerase II alpha (TOP2A), or more rarely phosphatase and tensin homolog (PTEN) have also been related to MpBC development [17,18,29,31,56]. Activating mutations of the phosphoinositide 3-kinase (PI3K) pathway and loss of the PTEN antagonist alter the microenvironment of the tumor and have been associated with resistance to therapy [38]. Furthermore, activation of the Wnt/β-catenin signaling pathway, which has been associated with dysregulated immune responses and cell cycle disruption, has been observed in MpBC cases [41].

According to the proteomic analysis performed by Djomehri et al., each MpBC subtype appears to have unique and active differentiation programs. However, they observed accumulated epithelial-to-mesenchymal transition and extracellular matrix production in MpBC cases compered to triple-negative breast cancers [29]. Additionally, Baum et al. reported that recurrent mutations in the HRAS and PIK3CA genes were related to activation of downstream pathways leading to excessive cell proliferation and tumor growth, especially in spindle cell carcinoma [17]. These molecular findings suggested not only a novel pathway of carcinogenesis but also potential therapeutic targets specific to each MpBC pathological subtype.

### 4.4. Clinicopathological Features

Despite the fact that the clinicopathological features of MpBC are similar to those of other high-grade breast carcinomas of no special type, metaplastic breast carcinoma is usually more aggressive than pure invasive ductal or invasive lobular cancers. It mainly affects post-menopausal women; however, cases in younger women have also been reported [6,19]. In our study, the mean age at diagnosis was 55 (±14.7 SD) years, while seven women younger than 45 years old were diagnosed with MpBC and treated in our institution and 44 women with MpBC younger than 45 years old were described in the literature. This neoplasm is often larger in size and presents as a rapidly growing palpable breast mass, with ill-defined borders on mammography, ultrasonography, and magnetic resonance imaging, as well as findings of invasive carcinoma [14]. Unfortunately, no specific radiological findings exist [34]. Mammography typically shows areas of a high-density mass with variable margins and no macrocalcifications, while ultrasound may show a microlobulated mass with complex echogenicity and solid and cystic components corresponding to necrosis and cystic degeneration [49,68]. On the other hand, MRI mammography, specifically T2-weighted imaging, shows a high-signal-intensity mass with neoplastic intensification similar to invasive breast carcinoma [68].

It is intriguing that invasion of the axillary lymph nodes is infrequent, while hematogenous metastasis occurs more often in cases of metaplastic carcinoma. MpBC shows a tendency for early hematogenous spread to distant organs such as the lungs, liver, and bones, while local recurrence is also quite frequent [9]. On the contrary, the rates of axillary metastasis may vary depending on tumor morphology. The hematogenous spreading route is particularly more common in subtypes with predominant sarcomatoid carcinoma in the spectrum [9,14,26]. Acar et al. reported that the risk of distant metastasis was higher in MpBC patients than in patients with ductal or lobular adenocarcinomas, while the risk of lymph node involvement was lower in MpBC patients [11].

### 4.5. Diagnosis of MpBC

It is well established that core needle biopsy is the gold standard method for the differential diagnosis of breast lesions, irrespective of breast tumor pathology, with both high sensitivity and specificity [80]. However, establishing the diagnosis of MpBC by fine needle aspiration (FNA) or needle core biopsy can be quite challenging, as the presence of metaplastic breast cancer cells can co-exist with poorly differentiated ductal breast carcinomas and rarely with other breast carcinoma types. The histopathological diagnosis of MpBC is based on the presence of clearly malignant cytological features associated with the presence of unusual cellular or stromal components. Therefore, the diagnosis of MpBC based on a small preoperative sample from a core needle biopsy can be elusive mainly because of the absence of standard cytological features of malignancy, either due to resemblance to benign breast lesions or due to the presence of poorly differentiated adenocarcinomas, as reported by Bataillon et al. [16].

### 4.6. Histopathology

Generally, breast carcinomas can be classified according to their microarray gene expression profiling. During the last decades, the use of DNA microarray and immunohistochemical methods has become quite popular in breast cancer research, altering the landscape of the standard of care by determining specific molecular subgroups of breast cancer, predicting prognosis, and guiding personalized therapy [2,6]. As far as metaplastic carcinoma is concerned, there is no pathognomonic pattern in the immunohistochemistry that is specific to MpBC diagnosis, although some characteristics prevail. Usually, MpBC does not express ER, PR, or HER2; thus, it can often be evaluated as a subgroup of triple-negative breast cancer (TNBC), considering its immunohistochemical characteristics [29]. However, its prognosis is worse than that of non-metaplastic TNBC. The plurality of MpBC is characterized as basal-like molecular subtypes according to the gene expression in microarrays [24,34,67]. Use of specific markers, such as antigens or antibodies, may facilitate the diagnosis of this type of breast cancer [34]. Various markers that can be easily detected by immunohistochemistry are expressed in the different histological subtypes of MpBC. For instance, specific markers such as cytokeratins and S100 protein present high sensitivity and specificity for spindle cell carcinoma [12]. It is of interest that the presence of different cell lines and thus distinct antigens, such as cytokeratins (AE1/AE3 and MNF116), basal cytokeratins (34βE12, CK5/6, CK14, and CK17), luminal cytokeratins (CK8/18, CK7, and CK19), and vimentin (mesenchymal cells), or myoepithelial cell markers (S-100 protein, actin, and high-molecular-weight cytokeratin) can establish the diagnosis of mixed metaplastic carcinoma [6,7,24,49].

In addition, myoepithelial markers, particularly p63, and EGFR are more frequently expressed in MpBC (approximately 70–80%) than in ductal carcinoma [7,20]. P63 has been proposed as a diagnostic marker for metaplastic carcinoma. The sensitivity and specificity of p63 for metaplastic breast carcinoma were 86.7% and 99.4%, respectively [46,49].

In addition to confirming the diagnosis, immunochemistry plays an important role in the sub-categorization of MpBCs as well as differentiation from other conditions, such as phyllodes tumor and pure sarcoma. A combination of molecular indicators, including myoepithelial and epithelial markers, has been proposed as a useful tool for confirming the diagnosis of metaplastic cancer in addition to the initial histopathological examination.

### 4.7. Treatment

Due to its rarity, the treatment guidelines for MpBC are still uncertain. Despite its worse prognosis and challenging treatment, there are no current, specific therapeutic guidelines for MpBC patients. Patients with MpBC are treated more aggressively than patients with ductal or lobular carcinoma, usually with mastectomy and adjuvant chemotherapy, in accordance with international guidelines depending on hormonal receptor and Her2 status and TNM stage [6]. MpBC patients usually present with a large breast mass, which indicates locally advanced disease, and typically patients are not candidates for breast-conserving surgery. Therefore, modified radical mastectomy or mastectomy with or without axillary dissection can be implemented [32,81]. The role of neoadjuvant chemotherapy in this setting remains unclear as MpBCs are relatively chemotherapy-refractory, especially compared to conventional triple-negative invasive breast cancers [24,32]. Henessy et al. reported a low rate of 10% for pathological complete response in patients with MpBCs following neoadjuvant chemotherapy [82]. Recently, the use of immunotherapy in the neoadjuvant setting has improved the rates of pathological complete response for triple-negative breast cancer [83]. It is of interest whether these results would also hold true for patients with metaplastic triple-negative carcinoma. Joneha et al. reported that PD-L1 was frequently overexpressed in MpBC, suggesting that patients with metaplastic carcinoma may benefit from immunotherapy [84].

As far as systemic therapy is concerned, several studies emphasized that the choice of appropriate chemotherapy regime depends on the histological type of metaplastic breast cancer. Chen et al. reported a modest response to taxane-based therapy [25]. Moreover, cases with a squamous epithelial component showed good response to cisplatin-based chemotherapy regimens, while cases with sarcomatous elements responded to doxorubicin- and ifosfamide-based regimens [7,22,26,32]. The intertumoral heterogeneity of the disease could explain the resistance to conventional chemotherapy [31,85].

On the other hand, targeted therapies have not been adequately studied for MpBC. As previously mentioned, BRCA mutations have been noted in MpBC cases, thus this group of patients could potentially benefit from treatment with poly (ADP-ribose) polymerase inhibitors, which are approved for patients with germline BRCA-mutated Her2-negative breast cancer [86,87].

Several studies reported mutations in the *PIK3CA* gene in patients with metaplastic carcinoma, implying a potential role for PI3K inhibitors in these patients [29,88]. Concurrently, patients with mutations in the PI3K/AKT/mTOR pathway may also benefit from mTOR inhibitors. Basho et al. treated patients with triple-negative breast cancer, including 59 patients with metastatic MpBC, with liposomal doxorubicin, bevacizumab, and an mTOR inhibitor, namely everolimus or temsirolimus, and reported better outcomes in MpBC cases than in non-metaplastic triple-negative tumors. Especially, patients with mutations in the PI3K pathway achieved complete responses, thus the role of mTOR inhibition in this setting should be furtherly studied [89].

Concurrently, adjuvant radiotherapy is recommended for MpBC patients, just as for patients with ductal carcinoma, and improved overall and disease-specific survival have been reported [36,72]. Hu et al. analyzed data from 1665 patients with metaplastic carcinoma in the Surveillance, Epidemiology, and End Results database and showed that adjuvant radiotherapy improved disease-related survival, especially in patients with triple-negative disease [85].

This systematic review included studies that reported MpBC cases as well as 46 cases treated in our institution in order to provide a synopsis of MpBC characteristics and prognosis. Our analysis showed that metaplastic carcinoma was more often diagnosed in post-menopausal women, at TNM stage II, and it did not usually express hormone receptors or Her2. Almost all patients included in our analysis underwent surgery, either mastectomy or breast-conserving surgery, while most of them also received chemotherapy as part of adjuvant, neoadjuvant, or systemic treatment. Some of them were also treated with radiotherapy. Our results revealed that surgical treatment and TNM stage were independent risk factors concerning patients’ overall survival. However, most of the patients included in our analysis underwent surgery, as this is the current standard of care for non-metastatic cases and only three cases with de novo metastatic cancer were not candidates for surgery. According to the literature, the factors associated with the prognosis of patients were age at diagnosis, tumor size, histopathologic subtype, tumor grade, TNM stage, and hormone receptor status. The histopathologic subtype may play an important role for patients’ prognosis. For instance, several studies have associated spindle cell carcinoma with worse prognosis than matrix-producing and squamous carcinomas [18]. However, our analysis did not reveal any correlation between histological subtype and prognosis. This finding was in accordance with the results from three large studies analyzing data from the Surveillance, Epidemiology, and End Results database that were recently published, which reported the biological characteristics and treatment options for MpBC and their associations with patients’ survival [76,85,90]. All three studies reported baseline tumor characteristics similar to our results: triple-negative subtypes that were mostly diagnosed at TNM stage II. As far as prognosis is concerned, Mao et al. [76] showed that MpBC prognosis was associated with age, tumor grade, TNM stage, and surgical treatment, while Lan et al. included only patients who had undergone surgery and showed that age, race, tumor size, lymph node status, and radiation treatment were related to patients’ outcome [90]. Similar findings were also reported by Hu et al., who identified age, tumor size, lymph node status, and surgical treatment as risk factors for survival [88]. Tumor size and lymph node status are two basic parts of the TNM staging system. Our analysis revealed that advanced TNM stage was associated with worse overall survival, while surgical treatment improved overall survival and had a tendency to improve disease-free survival as well. It is of interest that age was not associated with either overall or disease-free survival in our results. This could be attributed to the fact that most patients in our analysis were older than 45 years of age.

This is the first systematic review of the literature including published cases of metaplastic breast cancer as well as cases treated in one institution. However, the study has some limitations. The reviewed studies could not capture every subtle factor, some of which may be critical for clinicians. In addition, other important details, such as immunohistochemistry after surgery, were not sufficiently recorded in the databases. Moreover, our study cannot assess the gene modules associated with these basal-like tumors and the family histories of patients. Consequently, these data were not available for the systematic review.

## 5. Conclusions

In summary, MpBC is a very rare tumor presenting unique features, both clinical and pathological, which differentiates it from ductal carcinoma of the breast. Although metaplastic carcinoma can be tricky to diagnose both on a clinical and histopathological basis, early diagnosis is critical in order to achieve better outcomes for our patients. The rarity and specific characteristics of this neoplasm require personalized treatments that should be widely discussed by a multidisciplinary team and the diagnosis should be confirmed by immunohistochemistry, which also plays an increasingly crucial role in the classification of subtypes. Better understanding of the molecular pathways involved in the development of this type of cancer will contribute to the development of individualized therapeutic options. However, as data about MpBCs are still quite limited and MpBC patients are not usually included in clinical trials, there is an unmet medical need for an international MpBC patient registry in order to identify patients’ responses to treatment options and improve their overall care and survival.

## Figures and Tables

**Figure 1 medsci-11-00035-f001:**
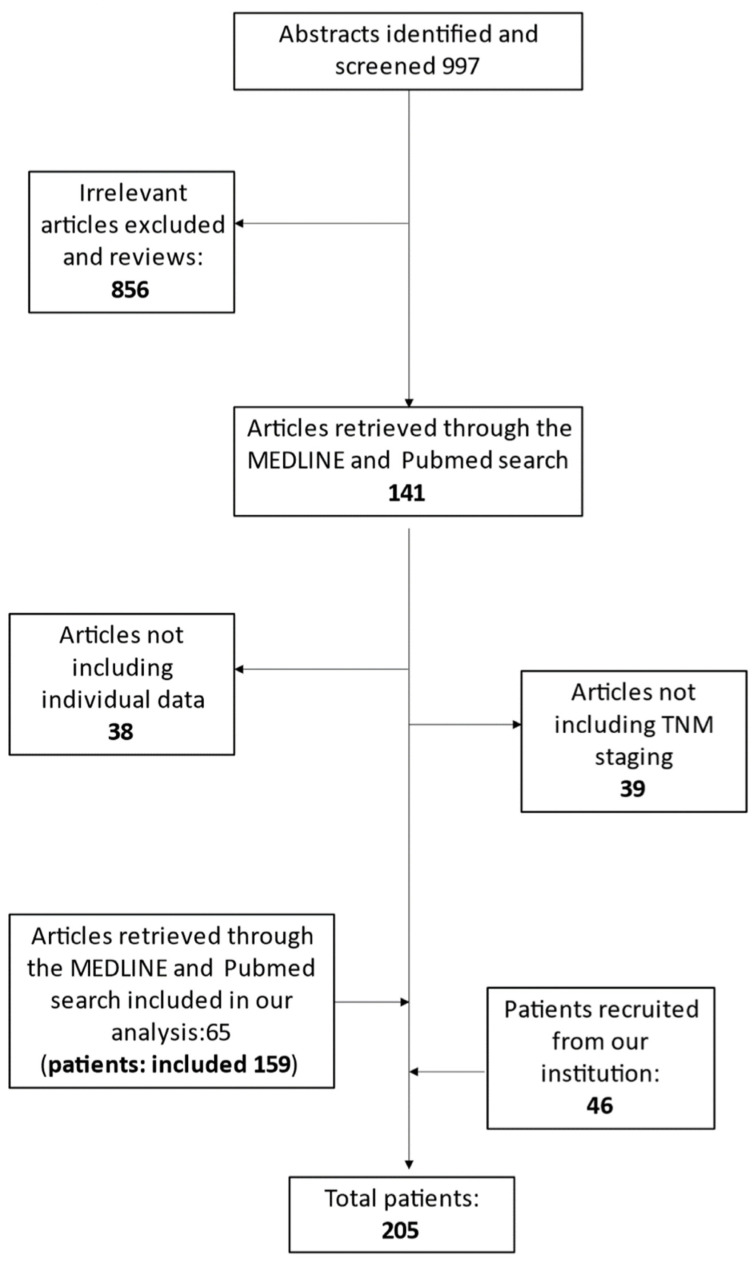
Search strategy.

**Figure 2 medsci-11-00035-f002:**
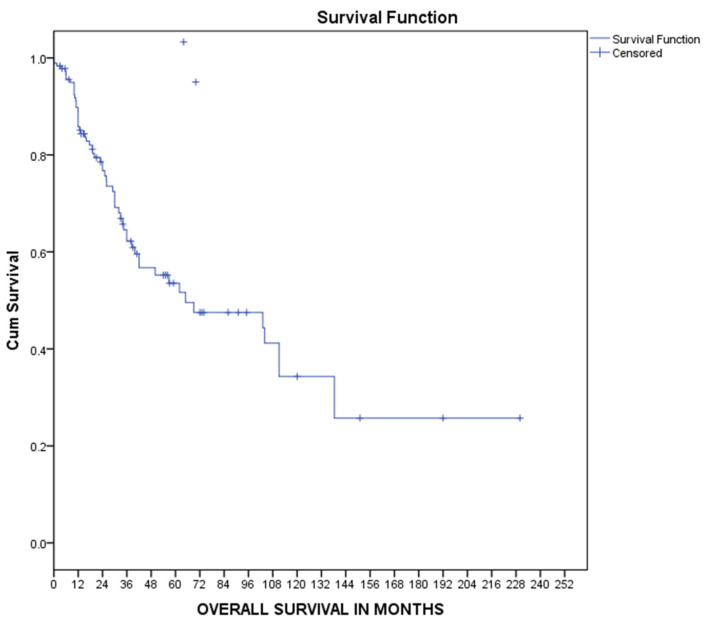
Kaplan–Meier overall survival curve.

**Figure 3 medsci-11-00035-f003:**
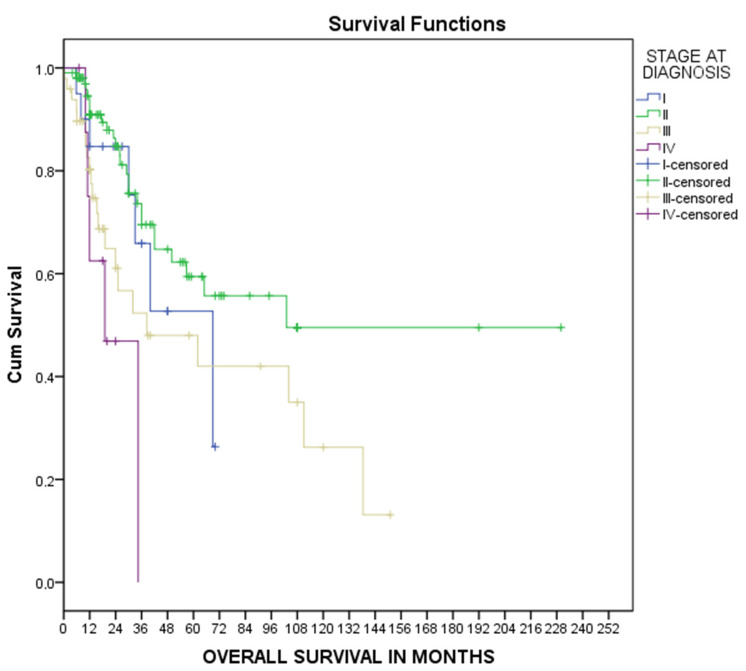
Kaplan–Meier overall survival curves for TNM stage.

**Figure 4 medsci-11-00035-f004:**
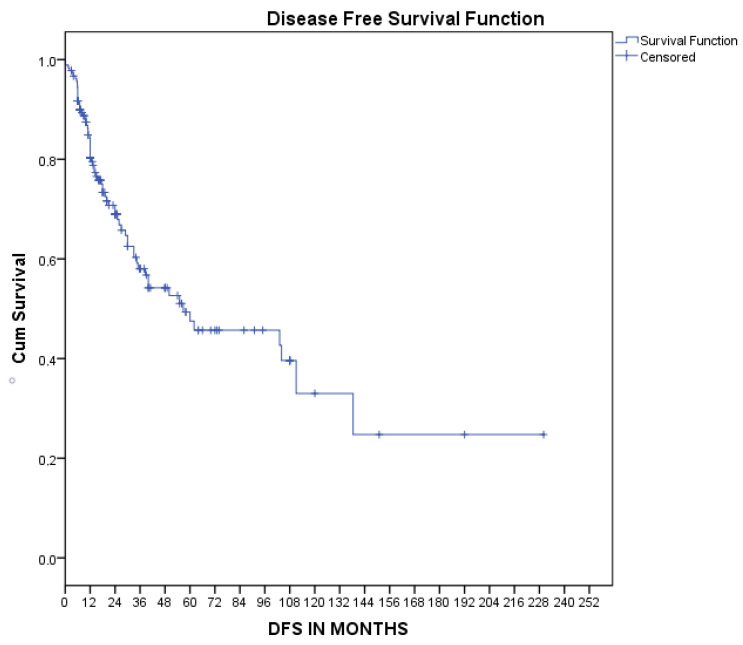
Kaplan–Meier disease-free survival curve.

**Table 1 medsci-11-00035-t001:** Mean and median baseline characteristics of 205 patients.

Age (years), mean (SD)	55 (14.7)
TNM stage, *n* (%)	
I	20 (9.8)
II	120 (58.5)
III	52(25.4)
IV	13 (6.3)
ER > 1% *n*/N (%)	19/180 (10.6)
PR > 1% *n*/N (%)	18/180 (10)
HER2 enriched *n*/N (%)	17/191 (8.9)
Ki 67 > 15% *n*/N (%)	32/40 (68.4)
Surgery *n*/N (%)	186/189 (98.4)
Chemotherapy *n*/N (%)	152/176 (86.4)
Radiation *n*/N (%)	88/154 (57.1)

**Table 2 medsci-11-00035-t002:** Cox regression analysis for association of patients’ baseline characteristic with overall survival.

	Univariate Cox Regression			Multivariate Cox Regression		
Patient Characteristic	HR	95% CI	*p*	HR	95% CI	*p*
Age	1	0.99–1.03	0.4	1	0.96–1.02	0.73
TNM stage	1.6	1.13–2.36	0.01	1.5	1.04–2.28	0.03
ER > 1%	1	0.43–2.35	0.99			
PR > 1%	1	0.43–2.35	0.99			
HER2 enriched	0.58	0.21–1.61	0.3			
Triple-negative phenotype	1.08	0.57–2.05	0.82			
Histology *	1.14	0.90–1.45	0.28			
Ki 67 > 15%	0.87	0.32–2.38	0.79			
Surgery	0.06	0.02–0.2	<0.001	0.11	0.02–0.54	0.01
Chemotherapy	0.68	0.32–1.38	0.23			
Radiation	1.02	0.63–2.28	0.58			

HR: hazard ratio, CI: confidence interval. * Data about mixed histology were only collected for patients treated in our hospital.

**Table 3 medsci-11-00035-t003:** Cox regression analysis for association of patients’ baseline characteristics with disease-free survival.

	Univariate Cox Regression			Multivariate Cox Regression		
Patient Characteristic	HR	95% CI	*p*	HR	95% CI	*p*
Age	1.01	0.99–1.04	1.17	1.01	0.98–1.03	0.59
TNM stage	1.3	0.85–2	0.23	1.36	0.84–2.21	0.21
ER > 1%	0.42	0.11–1.73	0.23			
PR > 1%	0.69	0.21–2.24	0.53			
HER2 enriched	0.41	0.1–1.7	0.22			
Triple-negative phenotype	2.08	0.81–5.33	0.13			
Histology *	1.15	0.90–1.46	0.26			
Ki 67 > 15%	0.82	0.31–2.18	0.69			
Surgery	0.05	0.01–0.23	<0.001	0.17	0.98–1.03	0.06
Chemotherapy	0.45	0.21–0.94	0.03	0.19	0.23–1.34	0.55
Radiation	1.72	0.87–3.4	0.12			

HR: hazard ratio, CI: confidence interval. * Data about mixed histology were only collected for patients treated in our hospital.

## Data Availability

Data are available upon request.

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
