# Peer review of "Metaplastic Carcinoma of the Breast: Case Series of a Single Institute and Review of the Literature"

_medsci, 2023, doi:10.3390/medsci11020035_

Round 1

Reviewer 1 Report

This is an original way to combine series with patient cases for a rare entity.

Comments. 

The abstract contains numbers in () that seems to reflect confidence intervals. It is not clear and it adds little value as it is not a unique serie, my recommendation is to delete from the abstract, and you will reflect in the results. A second aspect is about survival reporting data: Both OS and DFS are reported as means instead of medians and is not the natural way as the mean OS is shorter than the DFS. My recommendation is to report as medians or even better as 5 and/or 10 years OS and DFS rates. 

About the analysis and conclussions I have some concerns and recommendations. There is no analysis related to: 

1.- Molecular phenotype: Although in the table is clear that for some cases the ER/PR/HER2 status is missed, it will be very interesting to see if there are differences in behavior between the "triple negative" vs the others. 

2.- It could be also interesting to analyze if there are differences in OS/DFS by the histological subtype, particularly the ones with mixed subtype versus the more "pure" messenquimal-scquamous.

A third concern relates to the consideration of surgery as a mayor relevant factor for OS. In the tables there was only 3 cases without surgery and 2 of them were stage IV, so the correlation is strong. This aspect should be pointed out in the discussion  

Author Response

This is an original way to combine series with patient cases for a rare entity.

Comments. 

The abstract contains numbers in () that seems to reflect confidence intervals. It is not clear and it adds little value as it is not a unique serie, my recommendation is to delete from the abstract, and you will reflect in the results. A second aspect is about survival reporting data: Both OS and DFS are reported as means instead of medians and is not the natural way as the mean OS is shorter than the DFS. My recommendation is to report as medians or even better as 5 and/or 10 years OS and DFS rates. 

We thank the reviewer for his insightful remark. In our revised manuscript we have revised our statistical analysis and we report both OS and DFS as median values [IQR].

About the analysis and conclusions I have some concerns and recommendations. There is no analysis related to: 

1.- Molecular phenotype: Although in the table is clear that for some cases the ER/PR/HER2 status is missed, it will be very interesting to see if there are differences in behavior between the "triple negative" vs the others. 

We thank the reviewer for his remark. 78.5% of the cases with known hormanal and HER2 status were triple negative. However, Cox regression analysis did not reveal any correlation between the triple negative phenotype and either OS or DFS, as you can see in our revised Tables 3 and 4.

2.- It could be also interesting to analyze if there are differences in OS/DFS by the histological subtype, particularly the ones with mixed subtype versus the more "pure" messenquimal-scquamous.

We agree with the reviewer that histological subtypes of metaplastic breast carcinoma may play an important role on survival. However, most of the studies included in our analysis referred to the disease as “metaplastic carcinoma” without further classification.

Our analysis also included patients treated in our institution for whom data about histological subtypes were available. As shown in our revised Tables 3 and 4 Cox regression analysis did not reveal any correlation between the histological subtype and either OS or DFS.

A third concern relates to the consideration of surgery as a mayor relevant factor for OS. In the tables there was only 3 cases without surgery and 2 of them were stage IV, so the correlation is strong. This aspect should be pointed out in the discussion.

We thank the reviewer for this remark. We agree that the correlation is strong as per standard of care non metastatic patients will be treated with surgery. However, multivariate analysis including TNM stage confirmed surgical treatment as an independent risk factor for survival.

Reviewer 2 Report

I would like to thank the authors for the interesting article, that discusses the current role of metaplastic breast cancer. All in all, metaplastic breast cancers are rare (0.2-5% of breast cancer diagnosis). As these tumors tend to be more aggressive and mostly triple negative, treatment is often more difficult. The manuscript is a systematic review and included 46 patients who were treated in their home institution. All in all, data from 205 patients was included in the analysis.

Enclosed you find major suggestions for improvement:

1)    Line 47-52: Please add reference.

2)    Line 49: Here is one “t” in the sentence: “Reduced access to t healthcare …”

3)    Line 53-54: Please rephrase this sentence. It is not easy to understand what the author wants to say with this sentence.

4)    Line 57-60: Please add reference.

5)    Line 74 ff: The section of Materials and Methods is not precisely enough. For example, it is not known what data the authors extracted during the systematic review. It only says “TNM stadium”. Whereas in their own databank (with 46 patients being involved): tumor characteristics (e.g., ER/PR, Her2neu, Ki67, treatment) are obtained from medical records. Is it possible to specify how and which information the authors extracted?

6)    Line 97: Does disease-free survival also include progression-free survival in metastatic situation?

7)    Line 101, Statistics: Tests of normality, (e.g., the Kolmogorov–Smirnov test) should be performed to test for normal distribution. Only if variables are normally distributed, mean values + standard deviation is used; otherwise, it is recommended to present the values as median with range.

8)    What independent prognostic factors were tested in univariable Cox regression? (Line 105). Please add here (even if they are explained later).

9)    Line 112: How were the articles considered irrelevant? Two independent reviewers, etc.? Please explain.

10) In my opinion, table 1 does not provide any additional information and does not necessarily need to stay within the manuscript. The information on the histological type is already described in the manuscript.

11) Line 142-159: Why is the overall survival 100.6 (12.7) moths and the disease-free survival 147.49 (10.97) moths? Are you sure that there is no statistical error?? It seems unlikely that the overall survival is shorter than the disease-free survival… 

12) Line 146 ff.: Why was age included in the multivariate cox regression? Usually only the variables that are univariate statistically significant are included in the multivariable analysis (-so only TNM and surgery).

13) Line 162: Again, why did you include age and TNM in multivariate analysis when they were not statistical influencing factors in univariate analysis?? Please explain or correct the statistical analysis.

14) Could you give numbers about the influence of TNM on overall survival? (e.g., In stage I, median overall survival was xx months; in stage II, overall survival was xx months; and so on .) Furthermore, a Kaplan Meier graphic showing the survival curves depending on the TNM stages would also be very descriptive.

15) Line 182-185: According to World Health Organization classifications, MpBC can be devided … ïƒ Please ad reference.

16) Line 196-197: If the author looked for distant relapses, disease-free survival, or overall survival in different subtypes, please ad this information to the results section, as well.

17) Line 256-257: “invasion of the lymph nodes is infrequent while hematogenous metastasis occurs more often …” Please ad reference!

18) Line 267: FNA- Please explain abbreviations.

19) Line 267-269: Please ad reference.

20) Line 276-284: Please ad references.

21) Line 317: “relativel”- did you mean “relatively”?

22) In the present study, 98.4 % of all patients received surgery. It is therefore not surprising that surgery is associated with better overall survival. Those 1.6 % of patients that didn’t receive surgery might already have advanced disease and could not be operated? However, it would be very interesting to know if there is a distinction between the various surgical procedures: e.g., breast-conserving vs. mastectomy- is there any data on this? Does the type of surgery have an influence on overall or disease-free survival?

23) Line 363: You explain that the histopathological subtype is playing an important role in the prognosis of patients with MpBC. Why didn’t you include it in your analysis regarding overall and disease-free survival? Maybe the analysis could be added? 

24) The discussion section is very long in total. In order to make the discussion a little bit clearer, I would recommend inserting subheadings before the paragraphs. Furthermore, the discussion of the own data in the scientific context is very brief.

All in all, I think it is a current topic. But the manuscript needs a statistical revision in some places and, if possible, the inclusion of further analyses (histological subtypes, types of surgery). A Kaplan Meier graphic showing the survival curves depending on the TNM stages would also be very descriptive. Furthermore, the discussion section is very long and should be better structured. On the other hand, the discussion of your own data is very brief and should be more detailed put in context to current literature.

Author Response

I would like to thank the authors for the interesting article, that discusses the current role of metaplastic breast cancer. All in all, metaplastic breast cancers are rare (0.2-5% of breast cancer diagnosis). As these tumors tend to be more aggressive and mostly triple negative, treatment is often more difficult. The manuscript is a systematic review and included 46 patients who were treated in their home institution. All in all, data from 205 patients was included in the analysis.

We thank the reviewer for his remarks.

Enclosed you find major suggestions for improvement:

1)    Line 47-52: Please add reference. It is added in our revised manuscript.

2)    Line 49: Here is one “t” in the sentence: “Reduced access to t healthcare …”. It is added in our revised manuscript.

3)    Line 53-54: Please rephrase this sentence. It is not easy to understand what the author wants to say with this sentence.

We have rephrased the sentence in our revised manuscript

4)    Line 57-60: Please add reference. It is added in our revised manuscript.

5)    Line 74 ff: The section of Materials and Methods is not precisely enough. For example, it is not known what data the authors extracted during the systematic review. It only says “TNM stadium”. Whereas in their own databank (with 46 patients being involved): tumor characteristics (e.g., ER/PR, Her2neu, Ki67, treatment) are obtained from medical records. Is it possible to specify how and which information the authors extracted?

Data regarding the age, TNM stage, immuno-histochemical characteristics of tumor (ER, PR, HER2, ki67%), treatment approach (type of surgery, neoadjuvant or adjuvant treatment), and data related to the follow-up of the patients (overall survival, disease free survival).

6)    Line 97: Does disease-free survival also include progression-free survival in metastatic situation?

Yes.7)    Line 101, Statistics: Tests of normality, (e.g., the Kolmogorov–Smirnov test) should be performed to test for normal distribution. Only if variables are normally distributed, mean values + standard deviation is used; otherwise, it is recommended to present the values as median with range.

The Kolmogorov-Smirnov goodness-of-fit test as used to assess the distribution of each variable. We have updated the section “2.2. Statistical analysis” accordingly.

8)    What independent prognostic factors were tested in univariable Cox regression? (Line 105). Please add here (even if they are explained later).

Known prognostic risk factors such as age, TNM stage, ER, PR, HER2 and ki-67 expression, surgery, chemotherapy and radiation were tested in our original Cox regression analysis, as it was descripted in the results sections. In our revised manuscript, we have updated our analysis and included triple negative disease and mixed vs pure histological subtypes (in cases treated in our institution).

We have also added the above mentioned information in the Materials and methods section.    

9)    Line 112: How were the articles considered irrelevant? Two independent reviewers, etc.? Please explain.

We thank the reviewer for his remark. As explained in our revised manuscript two independent investigators, extracted data from each eligible study based on the title, abstract and full text utilizing a predefined form.

10) In my opinion, table 1 does not provide any additional information and does not necessarily need to stay within the manuscript. The information on the histological type is already described in the manuscript.

We have now moved the table to supplementary.

11) Line 142-159: Why is the overall survival 100.6 (12.7) moths and the disease-free survival 147.49 (10.97) moths? Are you sure that there is no statistical error?? It seems unlikely that the overall survival is shorter than the disease-free survival… 

We thank the reviewer for his insightful remark. In our revised manuscript we have revised our statistical analysis and we report both OS and DFS as median values [IQR].

12) Line 146 ff.: Why was age included in the multivariate cox regression? Usually only the variables that are univariate statistically significant are included in the multivariable analysis (-so only TNM and surgery).

All univariate statistically significant variables and potentially biological important variables were included in our multivariate analysis, as described in the 2.2. Statistical analysis section of our revised manuscript.  However, based on the reviewer’s remark, we performed multivariable analysis including only TNM stage and surgery, which yielded similar results.

13) Line 162: Again, why did you include age and TNM in multivariate analysis when they were not statistical influencing factors in univariate analysis?? Please explain or correct the statistical analysis.

Similarly to your previous comment, we would like to point out that in our multivariate analysis, we included all univariate statistically significant and potentially biological important variables, namely age and TNM stage. Once again, based on the reviewer’s remark we performed multivariable analysis including only chemotherapy and surgery, which yielded similar results.

14) Could you give numbers about the influence of TNM on overall survival? (e.g., In stage I, median overall survival was xx months; in stage II, overall survival was xx months; and so on .) Furthermore, a Kaplan Meier graphic showing the survival curves depending on the TNM stages would also be very descriptive.

We thank the reviewer for his insightful remark. In our revised manuscript we have included Kaplan Mayer curves for overall survival based on TNM status, as shown in the section 3.3 Overall Survival and in Figure 3, In our revised manuscript.

15) Line 182-185: According to World Health Organization classifications, MpBC can be devided … àPlease ad reference.

It is added in our revised manuscript.

16) Line 196-197: If the author looked for distant relapses, disease-free survival, or overall survival in different subtypes, please ad this information to the results section, as well.

We agree with the reviewer that histological subtypes of metaplastic breast carcinoma may play an important role on survival. However, most of the studies included in our analysis referred to the disease as “metaplastic carcinoma” without further classification.

Our analysis also included patients treated in our institution for whom data about histological subtypes were available and univariate Cox regression analysis did not reveal any correlation between the subtype and either OS or DFS, as shown in Tables 3 and 4.17) Line 256-257: “invasion of the lymph nodes is infrequent while hematogenous metastasis occurs more often …” Please ad reference!

It is added in our revised manuscript.

18) Line 267: FNA- Please explain abbreviations.

The abbreviation is explained in our revised manuscript.

19) Line 267-269: Please ad reference.

It is added in our revised manuscript.20) Line 276-284: Please ad references.

It is added in our revised manuscript.21) Line 317: “relativel”- did you mean “relatively”?

Corrected.

22) In the present study, 98.4 % of all patients received surgery. It is therefore not surprising that surgery is associated with better overall survival. Those 1.6 % of patients that didn’t receive surgery might already have advanced disease and could not be operated? However, it would be very interesting to know if there is a distinction between the various surgical procedures: e.g., breast-conserving vs. mastectomy- is there any data on this? Does the type of surgery have an influence on overall or disease-free survival?

We thank the reviewer for the remark. As per standard of care, non metastatic breast cancer patients are treated with surgery. However, multivariate analysis including TNM stage confirmed that surgical treatment is an independent risk factor.

Data regarding surgical procedures are not available for most published studies, therefore this analysis could not be performed.  

23) Line 363: You explain that the histopathological subtype is playing an important role in the prognosis of patients with MpBC. Why didn’t you include it in your analysis regarding overall and disease-free survival? Maybe the analysis could be added? 

As we mentioned in a previous comment data about histological subtype were not provided in most published studies. We have updated our results including Cox regression analysis for histological type which did not stastically affected OS and DFS.

In our revised manuscript we have also added a comment in the discussion about those findings which are in accordance with three large studies published recently. 

24) The discussion section is very long in total. In order to make the discussion a little bit clearer, I would recommend inserting subheadings before the paragraphs. Furthermore, the discussion of the own data in the scientific context is very brief.

We thank the reviewer for his remark. In our manuscript, we aimed at providing a synopsis on the current literature dealing with general aspects of the metaplastic carcinoma of the breast. In our revised manuscript, we have added subheadings through discussion. In order not to exceed the word limitation of the journal for reviews, we have indeed limited the discussion of our own results.  

All in all, I think it is a current topic. But the manuscript needs a statistical revision in some places and, if possible, the inclusion of further analyses (histological subtypes, types of surgery). A Kaplan Meier graphic showing the survival curves depending on the TNM stages would also be very descriptive. Furthermore, the discussion section is very long and should be better structured. On the other hand, the discussion of your own data is very brief and should be more detailed put in context to current literature.

Round 2

Reviewer 2 Report

Thank you for the revision of the manuscript. In my opinion, the authors tried to incorporate all suggestions for improvement. Nevertheless, I have still some major concerns.

1)    Why do patients with stage III breast cancer have a median OS of 15.4 months, while patients with stage IV have a median OS of 18 months? That makes no sense. It might be because of the low number of patients with 13 patients in stage IV breast cancer. Nevertheless, this should be mentioned and discussed in the discussion.

2)    Furthermore, I have concerns regarding the overall survival. Median overall survival of all patients is 66 months. But, stage I disease was 26 [25.5] months, for stage II disease 24 [38.25] months, for stage III 15.4 [25.25] and for metastatic disease 18 [11.5] months. How is that possible?? There must be an error in the calculation. With an median overall survival of 66 months, their should be at least one TNM group that has 66 months or more median OS to get this value of the calculation.

3)    I personally don’t understand why the OS is shorter than the DFS. Or, why the median DFS is not reached, but the overall survival is. That makes no sense. There has to be an error in the data, or there must be too many patients who were lost to follow-up etc. 

So, how many patients were lost to follow-up? For how many patients do you have data on DFS? What was the minimum time of observation for all patients? Did you use data from databases, or did you only extract the data from different manuscripts?

4)    186 from out of 189 patients (98.4%) received surgery. So, there are only 3 patients who didn’t receive surgery and these 3 patients suffered from de novo metastatic disease. For this reason, it is obvious that these 3 patients have worse overall and disease-free survival than the other patients. I find it very difficult to take this as a “core conclusion”.

5)    I have also one formal note in line 126: “informa_tion” ïƒ  please correct: “information”.

Author Response

Thank you for the revision of the manuscript. In my opinion, the authors tried to incorporate all suggestions for improvement. Nevertheless, I have still some major concerns.

 We thank the reviewer for his comments.

1)    Why do patients with stage III breast cancer have a median OS of 15.4 months, while patients with stage IV have a median OS of 18 months? That makes no sense. It might be because of the low number of patients with 13 patients in stage IV breast cancer. Nevertheless, this should be mentioned and discussed in the discussion.

We would like to thank the reviewer for raising this point. For overall analysis at the whole population we used censored analysis, which was not used at subgroup analysis based on stage. Censored subgroup analysis for TNM stage has been performed in our revised manuscript.

2)    Furthermore, I have concerns regarding the overall survival. Median overall survival of all patients is 66 months. But, stage I disease was 26 [25.5] months, for stage II disease 24 [38.25] months, for stage III 15.4 [25.25] and for metastatic disease 18 [11.5] months. How is that possible?? There must be an error in the calculation. With an median overall survival of 66 months, their should be at least one TNM group that has 66 months or more median OS to get this value of the calculation.

We would like to thank the reviewer for raising this point. Similarly to the previous comment for overall analysis at the whole population we used censored analysis, which was not used for subgroup analysis based on stage. Censored subgroup analysis for TNM stage was been performed in our revised manuscript.

3)    I personally don’t understand why the OS is shorter than the DFS. Or, why the median DFS is not reached, but the overall survival is. That makes no sense. There has to be an error in the data, or there must be too many patients who were lost to follow-up etc. 

So, how many patients were lost to follow-up? For how many patients do you have data on DFS? What was the minimum time of observation for all patients? Did you use data from databases, or did you only extract the data from different manuscripts?

We would like to thank the reviewer for raising this interesting remark. Our analysis included both published cases of metaplastic carcinoma and cases treated in our institution. The follow up period differed in published cases, so we performed censored survival analysis with the abovementioned results. 181 patients were included in overall survival analysis and 177 in disease free survival analysis respectively.

However, same patients included in our analysis have died without presenting with disease recurrence. In our updated analysis, all deceased patients are marked as recurrent.

4)    186 from out of 189 patients (98.4%) received surgery. So, there are only 3 patients who didn’t receive surgery and these 3 patients suffered from de novo metastatic disease. For this reason, it is obvious that these 3 patients have worse overall and disease-free survival than the other patients. I find it very difficult to take this as a “core conclusion”.

We agree with the reviewer that de novo metastatic patients who per standard of care did not receive surgery will have worse overall and disease free survival. This is pointed out in the discussion in our manuscript (lines 406-409). This is indeed a limitation of our study since we did not have more patients with de novo metastatic disease.

However, our results are in accordance with recent studies by Mao et al. and Hu et al. have identified surgical treatment as a risk factor for prognosis in patients with metaplastic breast cancer (lines 423-428).  

5)    I have also one formal note in line 126: “informa_tion” à please correct: “information”.

Corrected.

Round 3

Reviewer 2 Report

All proposed revisions were implemented accordingly.